# An Efficient and Reliable Tolerance-Based Algorithm for Principal Component Analysis

## Abstract

Principal component analysis (PCA) is an important method for dimensionality reduction in data science and machine learning. But, it is expensive for large matrices when only a few principal components are needed. Existing fast PCA algorithms typically assume the user will supply the number of components needed, but in practice, they may not know this number beforehand. Thus, it is important to have fast PCA algorithms depending on a tolerance. For $m \times n$ matrices where a few principal components explain most of the variance in the data, we develop one such algorithm that runs in $O(mnl)$ time, where $l \ll \min(m, n)$ is a small multiple of the number of principal components. We provide approximation error bounds that are within a constant factor away from optimal and demonstrate its utility with data from a variety of applications.

## 1 Introduction

### 1.1 The truncated SVD and PCA

Let $A$ be an $m \times n$ real matrix and $A = U\Sigma V^T$ its singular value decomposition (SVD). The rank-$k$ truncated SVD (rank-$k$ TSVD) of $A$ is the matrix $A_k := U_k \Sigma_k V_K^T$, where $U_k$ and $V_k$ are the first $k$ columns of $U$ and $V$, respectively, and $\Sigma_k$ is the leading $k \times k$ block of $\Sigma$. The columns $u_j$ and $v_j$ of $U_k$ and $V_k$ are the left and right singular vectors of $A$, respectively, and the diagonal entries $\sigma_1(A) \geq \cdots \geq \sigma_k(A) \geq 0$ are the singular values of $A$.

One common use of TSVD is in principal component analysis (PCA). This is a dimensionality reduction technique that aims to find the directions in which the data varies the most. It turns out that these directions are given by the top $k$ right singular vectors $v_j$, $1 \leq j \leq k$. Projecting the original data onto these directions then transforms the data from a high-dimensional space into a lower-dimensional one. Typically, $k$ is chosen so that these principal directions explain a certain amount of variance in the data. For example, if the user wanted to explain 99% of the variance, they would choose $k$ so that $\sum_{i=1}^{k} \sigma_i(A)^2 / \sum_{i=1}^{n} \sigma_i(A)^2 \geq 0.99$. PCA is used to compress data in a variety of settings such as images, training data for machine learning algorithms, and even neural network weight matrices (Xue et al., 2013).

In this work, we will be interested in a slightly different version of PCA. Let $\varepsilon$ be a user-prescribed tolerance, and instead choose $k$ so that $\sigma_k(A) \geq \varepsilon \geq \sigma_{k+1}(A)$. Since the $i$th principal component explains $\sigma_i(A)^2 / \sum_{i=1}^{n} \sigma_i(A)^2$ fraction of the total variance, we are essentially ignoring any components that explain less than $\varepsilon^2 / \sum_{i=1}^{n} \sigma_i(A)^2$ fraction of the total variance. This can be interpreted as discarding principal components corresponding to noise, where $\varepsilon$ describes the size of the noise.

Despite its utility and importance, the main drawback of using TSVD for PCA is that it is expensive, especially when the user needs only the top few singular values/vectors. Thus, a large body of research has been devoted to finding faster ways of computing it without sacrificing too much accuracy.

### 1.2 Prior Work

The literature on fast, approximate TSVD algorithms typically assume the user knows what rank $k$ to use. Recent work uses randomization to reduce the run time while still maintaining a high level of

accuracy. See, for example, Rokhlin et al. (2010) and Halko et al. (2010) and the references therein for typical examples of these types of algorithms. In Musco & Musco (2015), the authors present a randomized algorithm based on block Krylov subspace methods to compute an approximate TSVD. For a matrix $A$, rank $k$, and tolerance $\varepsilon$, the algorithm produces a matrix $Z$ whose columns approximate the top $k$ left singular vectors of $A$ and such that $\left\|A - ZZ^T A\right\|_2 \leq (1 + \varepsilon) \left\|A - A_k\right\|_2$. They also prove stronger bounds on the quality of the singular vectors, which is important for PCA. This algorithm is especially suited to sparse matrices, which can be multiplied quickly.

In some cases, the user may not know ahead of time what $k$ to use, so it is useful to consider algorithms which accept a desired precision $\varepsilon$ as input rather than rank. Algorithms in this vein incrementally build a matrix $Q$ with orthogonal columns and another matrix $B$ until $\|A - QB\| < \varepsilon$. Typically, the number of columns of $Q$ (or the number of rows of $B$) is quite small so that $QB$ is a compact approximation of the original data $A$. An approximate TSVD of $A$ can then be produced from the SVD of $B$. Recent work again uses randomization to reduce the run time. We present a prototypical example of this style of algorithm in Algorithm 1 (Yu et al., 2018). See Halko et al. (2010) and Martinsson & Voronin (2016) for more examples. While these algorithms guarantee a small approximation error, there are no guarantees on the accuracy of the singular values or vectors. We will compare the accuracy for Algorithm 1 to the proposed algorithm in Experiments.

---

**Algorithm 1** The randQB_EI algorithm for the fixed-precision problem

---

**Input:** an $m \times n$ matrix $A$; desired accuracy tolerance $\varepsilon$; block size $b$
**Output:** $Q, B$ such that $\|A - QB\|_F < \varepsilon$
$Q = [\,]; B = [\,];$
$E = \|A\|_F^2$
**for** $i = 1, 2, 3, \ldots$ **do**
$\quad \Omega_i = \operatorname{randn}(n, b)$
$\quad Q_i = \operatorname{orth}(A\Omega_i - Q(B\Omega_i))$
$\quad Q_i = \operatorname{orth}(Q_i - Q(Q^T Q_i))$
$\quad B_i = Q_i^T A$
$\quad Q = [Q, Q_i]$
$\quad B = \begin{bmatrix} B \\ B_i \end{bmatrix}$
$\quad E = E - \|B_i\|_F^2$
$\quad$ **if** $E < \varepsilon^2$ **then stop**
**end for**

---

### 1.3 OUR WORK

In this work, we propose an algorithm that, for a matrix $A$, accuracy tolerance $\delta$, and singular value tolerance $\varepsilon$, produces an approximate TSVD $\tilde{A}_k$ satisfying:

1. The rank $k$ of $\tilde{A}_k$ does not exceed the true rank of $A$, determined by the tolerance $\varepsilon$ as described above,

2. $\sigma_j(\tilde{A}_k) \geq (1 - \delta)\sigma_j(A)$ for $1 \leq j \leq k$,

3. $\left\|A - \tilde{A}_k\right\|_2 \leq \frac{1+\delta}{1-\delta}\varepsilon \approx (1 + 2\delta)\varepsilon$, and

4. If $k$ coincides with the true rank, then $\left\|A - \tilde{A}_k\right\|_2 \leq (1+\delta)\sigma_{k+1}(A) = (1+\delta)\left\|A - A_k\right\|_2$, i.e. the truncation error is a factor of $1 + \delta$ from optimal.

These properties are verified in the Appendix. The algorithm thus yields a high-quality approximation to TSVD and can be used in applications as an approximate PCA. The algorithm is fast for matrices whose singular values decay quickly when $\varepsilon$ is set so that $k$ will be relatively small.

## 2 PRELIMINARIES

Unless otherwise stated, we will consider matrices with more rows than columns. For a matrix with more columns than rows, apply the algorithm to its transpose.

### 2.1 FLIP-FLOP SPECTRUM REVEALING QR

The proposed algorithm is essentially a tolerance-based version of Flip-Flop Spectrum Revealing QR (FFQR) (Feng et al., 2019). For a matrix $A$ and integers $k \leq l$, FFQR produces an approximation to the rank-$k$ TSVD $A_k$ whose accuracy depends on the ratio $\sigma_{k+1}(A)/\sigma_{l+1}(A)$. Thus, if $A$ has rapidly decaying singular values, FFQR will be close to TSVD.

FFQR is computed as follows. Let $A$ be an $m \times n$ matrix ($m \geq n$) and $k \leq l$. Perform $l$ steps of Randomized QR with Column Pivoting (RQRCP) (Duersch & Gu, 2017) to get the factorization

$$A\Pi = QR = Q \begin{pmatrix} R_{11} & R_{12} \\ 0 & R_{22} \end{pmatrix},$$

where $\Pi$ is an $n \times n$ permutation matrix, $Q$ is an $m \times m$ orthogonal matrix, $R$ is $m \times n$, and $R_{11}$ is an $l \times l$ upper triangular matrix.

The next phase of FFQR involves performing extra "spectrum-revealing" column swaps on $R$ and using Givens rotations to restore its upper trapezoidal form. These swaps ensure $\|R_{22}\|_2 = O(\sigma_l(A))$. See Xiao et al. (2017) for more details.

Next, perform $l$ steps of QR on $R^T$ to get

$$R^T = PL^T = (P_1 \quad P_2) \begin{pmatrix} L_{11} & 0 \\ L_{21} & L_{22} \end{pmatrix}^T,$$

where $P$ is an $n \times n$ orthogonal matrix, $P_1$ is its leading $l$ columns, $L$ is an $m \times n$ matrix, and $L_{11}$ is $l \times l$ lower triangular. Putting the above together yields

$$A = QR\Pi^T = Q \begin{pmatrix} L_{11} & 0 \\ L_{21} & L_{22} \end{pmatrix} P^T \Pi^T.$$

Discard $L_{22}$ (as in truncated QRCP) and approximate $\begin{pmatrix} L_{11} \\ L_{21} \end{pmatrix}$ with its rank-$k$ TSVD $\hat{U}_k \hat{\Sigma}_k \hat{V}_k^T$:

$$Q \begin{pmatrix} L_{11} & 0 \\ L_{21} & L_{22} \end{pmatrix} P^T \Pi^T \approx Q \begin{pmatrix} L_{11} \\ L_{21} \end{pmatrix} P_1^T \Pi^T \approx Q(\hat{U}_k \hat{\Sigma}_k \hat{V}_k^T) P_1^T \Pi^T$$

Setting $\tilde{U}_k := Q\hat{U}_k$, $\tilde{\Sigma}_k := \hat{\Sigma}_k$, $\tilde{V}_k := \Pi P_1 \hat{V}_k$ gives the rank-$k$ approximation $A \approx \tilde{U}_k \tilde{\Sigma}_k \tilde{V}_k^T$.

In Feng et al. (2019), the authors prove the following bounds for FFQR. Given $\varepsilon > 0$ and $g > 1$, there are matrix-dependent quantities $g_1 \leq \sqrt{\frac{1+\varepsilon}{1-\varepsilon}}$, $g_2 \leq g$, $\tau \leq g_1 g_2 \sqrt{(l+1)(n-l)}$, and $\hat{\tau} \leq g_1 g_2 \sqrt{l(n-l)}$ such that for $1 \leq j \leq k$,

$$\sigma_j(\Sigma_k) \geq \frac{\sigma_j(A)}{\sqrt[4]{1 + \min\left(2\hat{\tau}^4, \tau^4(2 + 4\hat{\tau}^4)\left(\frac{\sigma_{l+1}(A)}{\sigma_j(A)}\right)^4\right)}}$$

and

$$\left\| A - \tilde{U}_k \tilde{\Sigma}_k \tilde{V}_k^T \right\|_2 \leq \sigma_{k+1}(A) \sqrt[4]{1 + 2\tau^4 \left(\frac{\sigma_{l+1}(A)}{\sigma_{k+1}(A)}\right)^4}.$$

### 2.2 THE QLP DECOMPOSITION

The basis of FFQR is the QLP decomposition (Stewart, 1999). Let $A$ be an $m \times n$ matrix. Perform QRCP on $A$ to obtain $A\Pi = QR$ and then perform QRCP on $R^T$ to get $R^T \Pi_1 = PL^T$, where

$L$ is lower triangular. Putting these together yields $A = Q\Pi_1 L P^T \Pi^T$. This is the pivoted QLP decomposition of $A$. Stewart observed that the diagonal entries $L_{ii}$ of $L$ closely track the singular values of $A$.

For the proposed algorithm, we choose not to pivot when factoring $R^T$. In this case, $L_{ii}$ will not track $\sigma_i(A)$ as well. We will discuss a partial remedy for this below. The advantage of not pivoting is that, just as in FFQR, we do not have to finish computing $R$ before computing $L$. Once we have performed $l$ steps of QRCP on $A$, we can compute the first $l$ rows of $L$ and thus have access to its first $l$ diagonal entries.

# 3 A Fast, Approximate, Tolerance-Based PCA Algorithm

## 3.1 Blocked FFQR

Since we do not know what $l$ is beforehand, we compute $R$ and $L$ incrementally in blocks. Select a block size $b$ and perform $b$ steps of RQRCP to get

$$A\Pi_1 = Q_1 \begin{pmatrix} R_{11}^{(b)} & R_{12}^{(b)} \\ 0 & R_{22}^{(b)} \end{pmatrix},$$

where $R_{11}^{(b)}$ is $b \times b$ upper triangular. The first $b$ rows of $R$ are essentially done since subsequent steps of RQRCP will only permute the columns of $R_{12}^{(b)}$. Perform QR on them (to keep the notation simple, we write this as an LQ factorization):

$$\begin{pmatrix} R_{11}^{(b)} & R_{12}^{(b)} \end{pmatrix} = \begin{pmatrix} L_{11} & 0 \end{pmatrix} P_1^T,$$

where $L_{11}$ is $b \times b$ lower triangular, and $P_1$ is orthogonal. We have just computed the first $b$ rows of $L$ and know the first $b$ diagonal entries. For the next block, continue RQRCP for another $b$ steps. The permutation matrix $\Pi_2$ in this block will affect only columns $b+1$ through $n$, leaving the first $b$ columns untouched. Thus, $\Pi_2$ can be written in block form as $\Pi_2 = \begin{pmatrix} I_b & 0 \\ 0 & \tilde{\Pi}_2 \end{pmatrix}$, where $I_b$ is the $b \times b$ identity matrix and $\tilde{\Pi}_2$ is an $(n-b) \times (n-b)$ permutation matrix. We now have

$$A\Pi_1\Pi_2 = Q_2 Q_1 \left( \begin{array}{c|cc} R_{11}^{(b)} & R_{12}^{(b)}\tilde{\Pi}_2 & \\ \hline 0 & R_{11}^{(2b)} & R_{12}^{(2b)} \\ 0 & 0 & R_{22}^{(2b)} \end{array} \right),$$

where $R_{11}^{(2b)}$ is $b \times b$ upper triangular. Since the first $b$ rows have changed, we must account for this in the previous LQ:

$$\begin{pmatrix} R_{11}^{(b)} & R_{12}^{(b)}\tilde{\Pi}_2 \end{pmatrix} = \begin{pmatrix} R_{11}^{(b)} & R_{12}^{(b)} \end{pmatrix} \Pi_2 = \begin{pmatrix} L_{11} & 0 \end{pmatrix} P_1^T \Pi_2.$$

Now apply the matrix $\Pi_2^T P_1$ to the newly completed rows $\begin{pmatrix} 0 & R_{11}^{(2b)} & R_{12}^{(2b)} \end{pmatrix}$ and perform LQ on the last $n-b$ columns to get

$$\begin{pmatrix} 0 & R_{11}^{(2b)} & R_{12}^{(2b)} \end{pmatrix} \Pi_2^T P_1 = \begin{pmatrix} L_{21} & L_{22} & 0 \end{pmatrix} P_2^T,$$

where $L_{22}$ is $b \times b$ lower triangular.

The orthogonal matrix $P_2$ affects only the last $n-b$ columns and can therefore be written in block form as $P_2 = \begin{pmatrix} I_b & 0 \\ 0 & \tilde{P}_2 \end{pmatrix}$. Hence, $\begin{pmatrix} L_{11} & 0 \end{pmatrix} = \begin{pmatrix} L_{11} & 0 \end{pmatrix} P_2^T$ and

$$\left( \begin{array}{c|cc} R_{11}^{(b)} & R_{12}^{(b)}\tilde{\Pi}_2 & \\ \hline 0 & R_{11}^{(2b)} & R_{12}^{(2b)} \end{array} \right) = \begin{pmatrix} L_{11} & 0 & 0 \\ L_{21} & L_{22} & 0 \end{pmatrix} P_2^T P_1^T \Pi_2,$$

showing that we have computed the first $2b$ rows of $L$. We can continue this procedure, computing $b$ rows of $L$ at a time. Once we decide to stop, we finish the remaining rows of $L$ by applying the

orthogonal matrices from all previous LQ factorizations to the last rows of $R$. For example, if we wanted to stop after 2 blocks, apply $\Pi_2^T P_1 P_2$ to $\begin{pmatrix} 0 & 0 & R_{22}^{(2b)} \end{pmatrix}$ to get

$$\begin{pmatrix} 0 & 0 & R_{22}^{(2b)} \end{pmatrix} \Pi_2^T P_1 P_2 = \begin{pmatrix} L_{31} & L_{32} & L_{33} \end{pmatrix}$$

and the partial QLP decomposition

$$A\Pi_1\Pi_2 = Q_2 Q_1 \left( \begin{array}{c|cc} R_{11}^{(b)} & R_{12}^{(b)}\tilde{\Pi}_2 \\ \hline 0 & R_{11}^{(2b)} & R_{12}^{(2b)} \\ 0 & 0 & R_{22}^{(2b)} \end{array} \right) = Q_2 Q_1 \begin{pmatrix} L_{11} & 0 & 0 \\ L_{21} & L_{22} & 0 \\ L_{31} & L_{32} & L_{33} \end{pmatrix} P_2^T P_1^T \Pi_2.$$

Afterwards, spectrum-revealing swaps can be performed if desired. For each swap and upper-trapezoidal restoration, some nonzero entries will appear above the diagonal in $L$. These are easily eliminated with Givens rotations.

## 3.2 Determining $l$

We now derive a criterion to determine when to stop factoring in blocked FFQR and to find $l$. Let $\varepsilon$ be the tolerance parameter, and define the rank $k$ by $\sigma_{k+1}(A) \leq \varepsilon \leq \sigma_k(A)$. One could use the bounds derived in Feng et al. (2019), using the diagonal entries of $L$ to estimate the singular values of $A$ and stopping when $\sigma_{l+1}(A)/\sigma_{k+1}(A)$ is sufficiently small. However, the dimension-dependent bounds for $\tau$ and $\hat{\tau}$ are impractical, so we will use a different bound.

In Feng et al. (2019), the authors prove that $\sigma_j(A)^4 \leq \sigma_j(\Sigma_k)^4 + 2\|R_{22}\|_2^4, 1 \leq j \leq k$. Rearranging this inequality gives

$$\sigma_j(\Sigma_k) \geq \sigma_j(A) \sqrt[4]{1 - 2\frac{\|R_{22}\|_2^4}{\sigma_j(A)^4}}, \quad 1 \leq j \leq k.$$

They also prove the following bound on the truncation error:

$$\left\| A - \tilde{U}_k \tilde{\Sigma}_k \tilde{V}_k^T \right\|_2 \leq \sigma_{k+1}(A) \sqrt[4]{1 + 2\frac{\|R_{22}\|_2^4}{\sigma_{k+1}(A)^4}}. \tag{1}$$

These bounds hold even without spectrum-revealing swaps. So, if $\|R_{22}\|_2 / \sigma_{k+1}(A)$, is small, then the leading $k$ singular values of $A$ will be revealed up to a certain number of digits and $\tilde{U}_k \tilde{\Sigma}_k \tilde{V}_k^T$ will be a nearly optimal rank-$k$ approximation. In practice, the above two bounds are sufficient because $\|R_{22}\|_2 = O(\sigma_l(A))$ already, without extra swaps. The earlier bounds still have theoretical value in that they show the algorithm works well when $A$ has rapidly decaying singular values.

The factors $\sqrt[4]{1 - 2\frac{\|R_{22}\|_2^4}{\sigma_j(A)^4}}$ and $\sqrt[4]{1 + 2\frac{\|R_{22}\|_2^4}{\sigma_{k+1}(A)^4}}$ are equal to $1 - \frac{1}{2}\frac{\|R_{22}\|_2^4}{\sigma_j(A)^4}$ and $1 + \frac{1}{2}\frac{\|R_{22}\|_2^4}{\sigma_{k+1}(A)^4}$, respectively, up to first order. Introduce an accuracy parameter $\delta$, and say we have $\frac{1}{2}\frac{\|R_{22}\|_2^4}{\sigma_{k+1}(A)^4} \leq \delta$. Then we have $\sigma_j(\tilde{\Sigma}_k) \geq \sigma_j(A)(1-\delta), 1 \leq j \leq k$, and $\left\| A - \tilde{U}_k \tilde{\Sigma}_k \tilde{V}_k^T \right\|_2 \leq \sigma_{k+1}(A)(1+\delta)$ up to first order. This means that $\approx -\log\delta$ digits of the top $k$ singular values of $A$ and optimal truncation error have been computed correctly. We can rewrite $\frac{1}{2}\frac{\|R_{22}\|_2^4}{\sigma_{k+1}(A)^4} \leq \delta$ as $\|R_{22}\|_2 \leq \sigma_{k+1}(A)\sqrt[4]{2\delta}$. This is the tolerance-based criterion to determine $l$.

### 3.2.1 Estimating $\sigma_{k+1}$ and $\|R_{22}\|_2$

To estimate $\|R_{22}\|_2$ and $\sigma_{k+1}(A)$ accurately, we use Stewart's observation that the diagonal entries $L_{ii}$ of $L$ closely track the singular values $\sigma_i(A)$ of $A$. As stated above, $L_{ii}$ will not track $\sigma_i(A)$ as well because we are not pivoting when factoring $R^T$. A partial remedy is simply to sort the $L_{ii}$'s. We show below that the resulting tracking behavior is similar in quality to that of fully pivoted QLP.

Let $L^{(j)}$ be the $j$-th largest diagonal entry of $L$ in magnitude, i.e. $\left|L^{(1)}\right| \geq \left|L^{(2)}\right| \geq \cdots \geq \left|L^{(n)}\right|$. In light of Stewart's observation, we will assume that there are constants $\alpha$ and $\beta$ such that $\alpha \left|L^{(j)}\right| \leq \sigma_j(A) \leq \beta \left|L^{(j)}\right|, 1 \leq j \leq n$. The values of $\alpha$ and $\beta$ will be estimated empirically below. A simple

way to interpret these inequalities is that for each diagonal entry $L_{jj}$, there is a singular value of $A$ in the interval $[\alpha \left| L_{jj} \right|, \beta \left| L_{jj} \right|]$. Consider $\{L_{jj} : \beta \left| L_{jj} \right| \leq \varepsilon\}$. This just corresponds to all the intervals $[\alpha \left| L_{jj} \right|, \beta \left| L_{jj} \right|]$ contained in $(-\infty, \varepsilon]$. For each $L_{jj}$ in the set, there is a singular value $\sigma_i(A)$ in the corresponding interval. Thus $\sigma_i(A) \leq \varepsilon$. Since $\sigma_{k+1}(A)$ is the largest singular value of $A$ less than or equal to $\varepsilon$, we must have $\sigma_i(A) \leq \sigma_{k+1}(A)$, which then implies $\alpha \left| L_{jj} \right| \leq \sigma_{k+1}(A)$. This yields a lower bound on $\sigma_{k+1}(A)$, namely $\max\{\alpha \left| L_{jj} \right| : \beta \left| L_{jj} \right| \leq \varepsilon\}$.

Since we will not know all the $L_{jj}$'s, we can obtain only a sub-optimal lower bound $s_{k+1}$ on $\sigma_{k+1}(A)$. Initialize $s_{k+1} = 0$. After $i$ blocks of blocked FFQR, update $s_{k+1} = \max\{\alpha \left| L_{jj} \right| : \beta \left| L_{jj} \right| \leq \varepsilon \text{ and } j \leq ib\}$.

To estimate $\|R_{22}\|_2 = \sigma_1(R_{22})$, we will use the first diagonal entry $L_{11}$ of $L$ in the fully pivoted QLP factorization. First, consider a general matrix $A$, and perform QRCP: $A\Pi = QR$. Then in fully pivoted QLP, $\left| L_{11} \right|$ is just the largest row norm of $R$. As noted in Stewart (1999), the largest row of $R$ is usually among the first few rows. Thus we can estimate $\|A\|_2$ using $\max\limits_{1 \leq i \leq q} \|R(i, :)\|_2$, for some small integer $q$. This requires only $q$ steps of QRCP.

We can apply this idea to estimate $\|R_{22}\|_2$. After $i$ steps of QRCP on $A$, the $R$ factor has the form $\begin{pmatrix} R_1^{(i)} \\ 0 \quad R_{22}^{(i)} \end{pmatrix}$, where $R_1^{(i)}$ is $i \times n$ upper triangular. The $R$ factor in the QRCP factorization of $R_{22}^{(i)}$ is just $R_1^{(n)}(i+1 : m, i+1 : n)$. Thus, QRCP-factoring $A$ automatically yields the QRCP factorizations of all the trailing blocks $R_{22}^{(i)}$. Using the 2-norm estimation scheme in the previous paragraph, after $j$ steps of QRCP, we have the upper bound $\left\|R_{22}^{(i)}\right\|_2 \leq \beta \max\limits_{i \leq \iota \leq i+q-1} \left\|R_1^{(j)}(\iota, :)\right\|_2 := \beta \left\|R_{22}^{(i)}\right\|_{j,q}$ for $1 \leq i \leq j - q + 1$.

Putting these estimates together will give us the final stopping criterion. After each block, we first update $s_{k+1}$ with the newly computed $L_{jj}$'s and then check if $\left\|R_{22}^{(i)}\right\|_{j,q} \leq \frac{1}{\beta} s_{k+1} \sqrt[4]{2\delta}$ for some $i$. The smallest $i$ for which this inequality holds will be $l + 1$.

---

**Algorithm 2** Approximate, tolerance-based PCA

---

**Inputs:** $A$, tolerance $\varepsilon$, accuracy $\delta$, block size $b$, number of rows $q$, oversampling size $p$ for RQRCP
**Outputs:** Rank $k$, $\tilde{U}_k$, $\tilde{\Sigma}_k$, $\tilde{V}_k$
$c \leftarrow 0, s_{k+1} \leftarrow 0$
**while** $c < n$ **do**
    Perform steps $c + 1$ to $c + b$ of RQRCP on $A$; update $Q$, $R$, and $\Pi$
    Compute rows $c + 1$ to $c + b$ of $L$; update $P$
    **for** $j = c + 1 : c + b$ **do**
        **if** $\left| L_{jj} \right| \leq \varepsilon/\beta$ **and** $\alpha \left| L_{jj} \right| \geq s_{k+1}$ **then**
            $s_{k+1} \leftarrow \alpha \left| L_{jj} \right|$
        **end if**
    **end for**
    **for** $i = 1 : c + b - q + 1$ **do**
        **if** $\left\|R_{22}^{(i)}\right\|_{c+b,q} \leq \frac{1}{\beta} s_{k+1} \sqrt[4]{2\delta}$ **then**
            $l \leftarrow i - 1$
            **exit while loop**
        **end if**
    **end for**
    $c \leftarrow c + b$
**end while**
Compute rows $c + b + 1$ to $m$ of $L$.
Compute TSVD $\hat{U}_k \hat{\Sigma}_k \hat{V}_k$ of $L(:, 1 : l)$, where $k$ satisfies $\sigma_k(L(:, 1 : l)) \geq \varepsilon \geq \sigma_{k+1}(L(:, 1 : l))$.
$\tilde{U}_k \leftarrow Q\hat{U}_k, \tilde{\Sigma}_k \leftarrow \hat{\Sigma}_k, \tilde{V}_k \leftarrow \Pi P_1 \hat{V}_k$

---

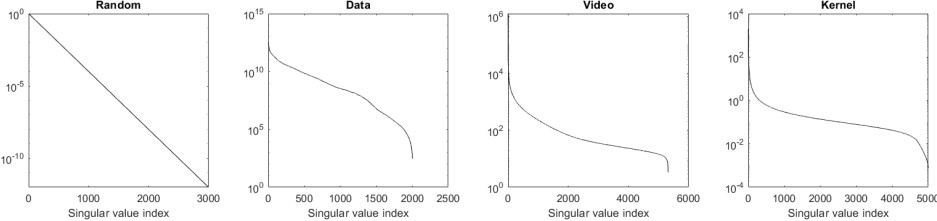

Figure 1: Singular value distributions of the test matrices

Table 1: $\alpha$ and $\beta$ values for the test matrices under the three schemes.

|  | Unpivoted | | Sorted | | Pivoted | |
|---|---|---|---|---|---|---|
|  | $\alpha$ | $\beta$ | $\alpha$ | $\beta$ | $\alpha$ | $\beta$ |
| Random | 0.710 | 1.39 | 0.742 | 1.33 | 0.745 | 1.32 |
| Data | 0.523 | 2.75 | 0.793 | 1.17 | 0.817 | 1.17 |
| Video | 0.321 | 2.85 | 0.838 | 1.85 | 0.840 | 1.58 |
| Kernel | 0.635 | 1.60 | 0.802 | 1.32 | 0.817 | 1.31 |

## 4 EXPERIMENTS

We use the following test matrices for our experiments:

1. A random $3000 \times 3000$ matrix with singular values decaying geometrically from 1 down to $10^{-12}$. We generate a $3000 \times 3000$ matrix with entries from a standard normal distribution, compute its SVD $U\Sigma V^T$, and replace the diagonal of $\Sigma$ with the desired singular value distribution.

2. A $2003 \times 2003$ data matrix (bcsstk13) from the SuiteSparse matrix collection (Davis & Hu, 2011). This matrix arises from a computational fluid dynamics problem. Its singular values decay from $\approx 10^{12}$ down to $\approx 10^2$.

3. A $19200 \times 5322$ matrix generated from a video from the UCF-Crime dataset (Sultani et al., 2018). The original video was a $240 \times 320$ RGB video consisting of 5322 frames. We resized the video by half to $120 \times 160$, converted it to grayscale, flattened each frame into a $19200 \times 1$ column vector, and then stacked these horizontally to form the final matrix.

4. A $5000 \times 5000$ kernel matrix generated from 5000 data points from the MNIST handwritten digits dataset (Lecun et al., 1998). We used the kernel function $k(x, x') = e^{-\gamma \|x - x'\|^2}$, where $\gamma = 1/(\text{median of pairwise distances between data points})^2$.

The singular values of each test matrix are plotted in Figure 1.

### 4.1 ESTIMATING $\alpha$ AND $\beta$

For each matrix $A$, we tested three singular value estimation schemes. See Table 1. For the first two columns ("Unpivoted"), we ran RQRCP to get $A\Pi = QR$ and then QR-factored $R^T = PL$. We recorded the minimum ($\alpha$) and maximum ($\beta$) values of $\sigma_i(A)/|L_{ii}|$. For the second two ("Sorted"), we recorded the minimum and maximum values of $\sigma_i(A)/|L^{(i)}|$, where $L^{(i)}$ is the $i$th largest diagonal entry of $L$ in magnitude. For the last two columns ("Pivoted"), we QRCP-factored $R^T\Pi_1 = PL$ and then recorded the minimum and maximum values of $\sigma_i(A)/|L_{ii}|$.

We observed that the tracking behavior can break down when $\sigma_i(A)$ is smaller than machine precision and thus ignored ratios corresponding to such $\sigma_i(A)$ when computing the minimum and maximum values. Therefore, it is recommended that the tolerance $\varepsilon$ be set at least a small factor above machine epsilon.

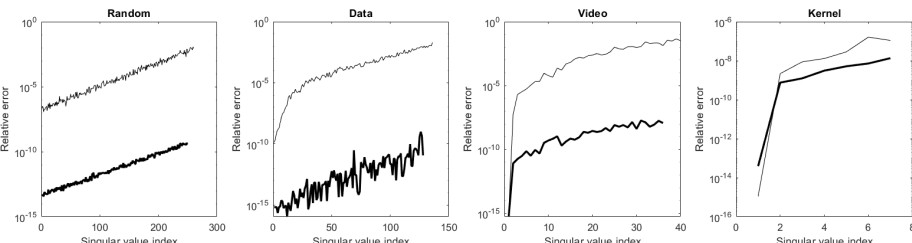

Figure 2: Relative singular value errors for the test matrices. Our algorithm is the bold line, randQB_EI is the thin one.

"Sorted" and "Pivoted" have similar $\alpha$ and $\beta$ values, with the latter slightly better overall, while "Unpivoted" tends to be worse than the other two. Based on the middle two columns, it seems that $\alpha \approx 0.7$ and $\beta \approx 2$ are reasonable values.

## 4.2 COMPARISON TO TSVD AND RANDQB_EI

Here we compare the proposed algorithm to TSVD and randQB_EI. Tests were coded in Fortran and run on a laptop with a 2.00 GHz Intel i7-4510U CPU with 16.0 GB of RAM.

First, we compare the proposed algorithm to TSVD. To compute the latter, the LAPACK routine dgesdd routine is used to compute the full SVD, which is then truncated based on the tolerance $\varepsilon$. For the random, data, and kernel matrices, tolerances corresponding to 99% explained variance are chosen. For the video matrix, we choose one corresponding to 99.9% explained variance because the first principal component already accounts for 99% of the variance.

It is not so simple to translate a Frobenius norm tolerance to a corresponding 2-norm tolerance, but we find that for matrices with geometrically decaying singular values, 99% explained variance roughly corresponds to a tolerance of $0.1 \left\| A \right\|_2$. The 2-norm of $A$ can be estimated after the first block of FFQR. For experimental purposes, we computed the SVD of each matrix and then selected the tolerance.

We set $\delta = 10^{-4}$ for all test matrices because machine learning algorithms typically only need a few digits of accuracy. But the larger singular values are computed with more accuracy because the accuracy of the $j$th singular value depends on $\left\| R_{22} \right\|_2 / \sigma_j(A)$. Finally, for all matrices, we set the block size $b = 64$, number of rows $q = 5$, and oversampling size $p = 5$.

The results are listed in Table 2. The proposed algorithm detects the rank $k$ correctly for each test matrix and is much faster than dgesdd. The column REOTE contains the relative error in the optimal truncation error $\left| \frac{\left\| A - \tilde{A}_k \right\|_2}{\left\| A - A_k \right\|_2} - 1 \right|$. This is always bounded above by $\delta$, but we see that in practice this relative error is much smaller.

Figure 2 plots the relative errors in the approximate singular values $\left| 1 - \frac{\sigma_j(\bar{\Sigma}_k)}{\sigma_j(A)} \right|$, $1 \leq j \leq k$ for each of the test matrices. These are again bounded by $\delta$. As expected, the larger singular values are computed more accurately. We also plot the relative errors in the singular values computed by randQB_EI, using block size 64; $\varepsilon = \sqrt{0.01} \left\| A \right\|_F$ for the random, data, and kernel matrices; and $\varepsilon = \sqrt{0.001} \left\| A \right\|_F$ for the video matrix. The authors of randQB_EI also include a power parameter $P$ in their implementation. We set $P = 1$ as in their paper.

Although we did not analyze the error in the singular vectors (in general, the error depends on the gap between the singular values), we compute the angles $\theta(v_j, \tilde{v}_j)$ between the right singular vectors and their approximations for both our algorithm and randQB_EI, and plot them in Figure 3. For our algorithm, the angles are all quite small, so the proposed algorithm finds good-quality approximations to the true singular vectors/principal directions.

Accuracy-wise, our algorithm performs better than randQB_EI, except on the kernel matrix. We found that setting $P = 0$ causes the accuracy of randQB_EI to drop below ours. The first few

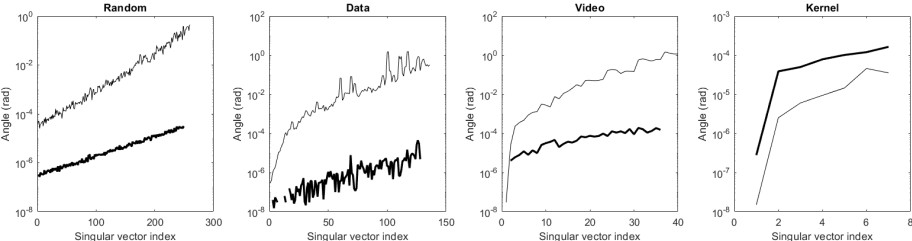

Figure 3: Angle between the top $k$ right singular vectors $v_j$ and their approximations $\tilde{v}_j$ for the test matrices. Our algorithm is the bold line, randQB_EI is the thin one.

Table 2: Comparison of the proposed algorithm to TSVD.

|  |  | dgesdd | | Proposed algorithm | | | | |
| --- | --- | --- | --- | --- | --- | --- | --- | --- |
| Matrix | $\varepsilon$ | $k$ | Time(s) | $l$ | $k$ | Time (s) | REOTE | Speed-up |
| Random | $1 \times 10^{-1}$ | 250 | 14.9 | 804 | 250 | 2.76 | $1.11 \times 10^{-16}$ | $5.39\times$ |
| Data | $1 \times 10^{11}$ | 128 | 4.41 | 759 | 128 | 1.38 | $2.22 \times 10^{-16}$ | $3.19\times$ |
| Video | $5.6 \times 10^{3}$ | 36 | 161 | 1692 | 36 | 50.2 | $9.55 \times 10^{-15}$ | $3.21\times$ |
| Kernel | $7.68 \times 10^{1}$ | 7 | 72.7 | 370 | 7 | 2.75 | $1.11 \times 10^{-15}$ | $26.4\times$ |

singular values of the kernel matrix are extremely large compared to the rest; thus, performing even just one power iteration effectively enhances the accuracy of randQB_EI on this matrix.

## 5 CONCLUSION

In this work, we developed an efficient algorithm for computing an approximate truncated SVD. In contrast to much of the literature, this algorithm truncates according to a tolerance rather than a fixed rank. We have also demonstrated that it provides high-quality approximations to both the singular values and vectors of the original matrix, thus making it suitable for use in applications as an approximate PCA.

ACKNOWLEDGMENTS

We would like to thank Jed Duersch for providing his code for RQRCP.

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

# A  APPENDIX

## A.1  VERIFICATION OF PROPERTIES 1-4

In this subsection, we denote the true rank as $k_{true}$, the rank detected by our algorithm as $\tilde{k}$, and the matrix output by our algorithm $\tilde{A}_{\tilde{k}}$. Recall $k_{true}$ is defined by $\sigma_{k_{true}+1}(A) \leq \varepsilon \leq \sigma_{k_{true}}(A)$. The detected rank $\tilde{k}$ is determined as follows. We run Blocked FFQR until the trailing block of $R$ is small enough and then take $l$ to be the smallest integer such that $\|R(l+1:m, l+1:n)\|_2 \leq \sigma_{k_{true}+1}(A)\sqrt[4]{2\delta}$. Denote $R(l+1:m, l+1:n)$ by $R_{22}$ for short. Afterwards, compute the SVD of $L_1 := L(:, 1:l)$ and define $\tilde{k}$ by $\sigma_{\tilde{k}+1}(L_1) \leq \varepsilon \leq \sigma_{\tilde{k}}(L_1)$.

First, note that $\tilde{k} \leq k_{true}$. To see this, first observe that $k_{true} = \#\{j : \sigma_j(A) > \varepsilon\}$. By the Cauchy Interlacing Theorem, $\sigma_j(L_1) \leq \sigma_j(A)$, $1 \leq j \leq l$. Thus we can only shift the singular values of $A$ downward, which will not increase the size of the above set. This proves Property 1.

For Property 2, it follows from computations in Feng et al. (2019) that $\sigma_j(A)^4 \leq \sigma_j^4(L_1) + 2\|R_{22}\|_2^4$, or

$$\sigma_j(L_1) \geq \sigma_j(A)\sqrt[4]{1 - 2\frac{\|R_{22}\|_2^4}{\sigma_j(A)^4}} \approx \sigma_j(A)\left(1 - \frac{1}{2}\frac{\|R_{22}\|_2^4}{\sigma_j(A)^4}\right), \quad 1 \leq j \leq l.$$

Plugging in $\|R_{22}\|_2 \leq \sigma_{k_{true}+1}(A)\sqrt[4]{2\delta}$ gives $\sigma_j(L_1) \geq \sigma_j(A)(1-\delta)$ for $1 \leq j \leq k_{true}+1$ and in particular for $1 \leq j \leq \tilde{k}$. This is Property 2.

For the last two properties, we refer to Equation 1. In the notation for this section, it reads:

$$\left\|A - \tilde{A}_{\tilde{k}}\right\|_2 \leq \sigma_{\tilde{k}+1}(A)\sqrt[4]{1 + 2\frac{\|R_{22}\|_2^4}{\sigma_{\tilde{k}+1}(A)^4}} \approx \sigma_{\tilde{k}+1}(A)\left(1 + \frac{1}{2}\frac{\|R_{22}\|_2^4}{\sigma_{\tilde{k}+1}(A)^4}\right).$$

Again, plugging in $\|R_{22}\|_2 \leq \sigma_{k_{true}+1}(A)\sqrt[4]{2\delta}$ and using the fact that $\tilde{k} + 1 \leq k_{true} + 1$ gives $\left\|A - \tilde{A}_{\tilde{k}}\right\|_2 \leq \sigma_{\tilde{k}+1}(A)(1+\delta)$. We see that if $\tilde{k} = k_{true}$, then this inequality is Property 4.

Finally, from the proof of Property 2 above, we have $\sigma_{\tilde{k}+1}(L_1) \geq \sigma_{\tilde{k}+1}(A)(1-\delta)$. Thus, $\sigma_{\tilde{k}+1}(A) \leq \frac{1}{1-\delta}\sigma_{\tilde{k}+1}(L_1)$ and $\left\| A - \tilde{A}_{\tilde{k}} \right\|_2 \leq \frac{1+\delta}{1-\delta}\sigma_{\tilde{k}+1}(L_1) \leq \frac{1+\delta}{1-\delta}\varepsilon$, which is Property 3.

Note that $\tilde{k} < k_{true}$ only when there are singular values slightly above the tolerance. The tolerance-based criterion ensures that up to first order $\sigma_j(L_1) \geq \sigma_j(A)(1-\delta)$. So only singular values satisfying $\sigma_j(A) \geq \varepsilon \geq \sigma_j(A)(1-\delta)$ can be perturbed below $\varepsilon$ and decrease the rank.

