# OpenReview forum: "An Efficient and Reliable Tolerance-Based Algorithm for Principal Component Analysis"
_ICLR.cc/2022/Conference — ICLR 2022 Submitted_

### Official Review · Reviewer_ps4E · 2021-10-29

**Correctness:** 4
**Technical Novelty And Significance:** 2
**Empirical Novelty And Significance:** 2
**Recommendation:** 5
**Confidence:** 3

**Main Review:**

Strengths: The paper is well written. The problem, all notations, and the derivation of the idea are all clearly presented, so the paper is easy to follow.

Weaknesses: The proposed methods seems to be a non-essential variation of the FFQR algorithm in Feng et al., (2019). While specifying $\varepsilon$ is more practical than specifying $k$ in practice and modifying the FFQR algorithm for accepting $\varepsilon$ requires tedious deviation, the overall contribution of the paper seems incremental. The main theoretical results are essentially the same as in Feng et al., (2019).

Minor suggestions:
- In Algorithm 1, it is better to make the meaning of 'randn()' and 'orth' more clear.
- Add a period at the ends of the third last displayed equation of page 3.
- Line 2 of page 4, change "Stewart" to "Stewart (1999)".


**Summary Of The Paper:**

This paper develops an algorithm that approximates the principal components corresponding to the top singular values. The algorithm requires the user specified tolerance instead of the number of top singular values, which is often the case in practice. It presents the detailed algorithm and evaluate its performance using numerical experiments.


**Summary Of The Review:**

It is a well written paper, but the contribution seems to be incremental.

---

### Official Review · Reviewer_ELuj · 2021-11-01

**Correctness:** 2
**Technical Novelty And Significance:** 3
**Empirical Novelty And Significance:** 3
**Recommendation:** 5
**Confidence:** 3

**Main Review:**

## Strengths
- PCA is an important problem that has attracted the attention of theoreticians and practitioners for many years. Accordingly, the paper is well motivated and certainly of interest to many.
- Parameter tuning (for sparsity level or regularization) is taken for granted but often relies on heuristics or solving a similar problem many times which is far from ideal. As such, a tolerance based approach with low time complexity is appealing and a valuable contribution.
- Their method seems to perform quite well compared to Algorithm 1 on realistic problems (examples 2-4) encountered in the world.
- The authors also do a nice job summarizing several of the methods their work is based on which lays a good foundation for their paper.

## Weaknesses
- There is a claim concerning the time complexity in the abstract but no further discussion. It would strengthen the paper if the authors presented their analysis.
- It is also unclear if the complexity analysis assumes access to the singular values of $A$, or to the matrix that tracks the singular values, i.e., $L$.
- In general, the paper provides mathematical reasoning, however the informal structure makes it difficult to follow arguments and whether or not the implemented algorithm satisfies the properties in 1.3 and what conditions are necessary for the stated time complexity.
- Although the method eliminates the need to provide a target rank, the proposed algorithm requires several other parameters, in particular $\alpha$, $\beta$, $b$, and $\epsilon$. From the discussion in 3.2.1, 4.1, and 4.2 these values must be considered well before choosing. It seems that we have traded one unknown parameter for several others.


## Questions/comments:
- Section 1.1
    - Should be lower case K for definition of TSVD
- Section 1.3
    - The "true rank" as defined in the appendix does not appear in the main text body (rank $k$ appears at the beginning of 3.2). This is confusing since true rank suggests number of singular values greater than zero...not greater than some threshold. I recommend using a different name.
    - Bullet point 4: if $k$ is the true rank of $k$, then $\sigma_{k+1}(A) = 0$, correct? Should this be $\sigma_{k+1}(\tilde A_k) = 0$? This seems to contradict bullet point 1 (maybe this is based on the alternative definition of rank?)
- Section 3.2.1
    - Do we need to assume the there exist such $\alpha$ and $\beta$? This would be true for a very small $\alpha$ and very large
- Section 4.2
    - How does runtime compare between randQB and the proposed method? A column in table 2 would be helpful in evaluating the methods merit.


**Summary Of The Paper:**

This paper presents a PCA algorithm that terminates after approximate singular values fall below a user provided threshold. The proposed method is based on the FFQR algorithm of Feng et al, but includes a tolerance based stopping criteria. It is stated that the algorithm runs $O(mnl)$ time. Experimental results are provided for comparison between TSVD and randQB_EI of Yue et al.

**Summary Of The Review:**

The problem is well motivated and of considerable interest to the community, the numerical experiments appear promising as well. On the other hand, formal proofs are missing as is any discussion of time complexity results which seems to be a central contribution. At the moment, I believe the paper is below the acceptance threshold.

---

### Official Review · Reviewer_oomW · 2021-11-02

**Correctness:** 3
**Technical Novelty And Significance:** 2
**Empirical Novelty And Significance:** 1
**Recommendation:** 3
**Confidence:** 3

**Main Review:**


There’s extensive work on scaling up PCA such as the randomized algorithm Halko et., al. (2010). In the prior work section, the authors argue that the existing randomized algorithms only guarantee a small approximation error, but not the accuracy of the computed singular values or space. I think the claim is not true as the accuracy of the singular values/space follows from classic results, e.g., the
Hoffman-Wielandt bound, in matrix perturbation theory.

The threshold parameter epsilon in the proposed algorithm is introduced to specify the noise level. However, this parameter may not be very useful as data is typically standardized. I think it is helpful to establish the connection between the parameter with the rank of the algorithm output.

The presentation and the experiments could be improved. This paper lacks necessary evaluation on real-life datasets. The authors report numerical results of the proposed algorithm on synthetic data. For a fair evaluation, I think it would be helpful to include the state-of-the-art randomized algorithms as well as the robust PCA as baselines.

**Summary Of The Paper:**

In this paper, the authors propose an approximate algorithm for PCA on large data. The proposed algorithm computes only the principle components associated with the singular values larger than a given threshold.

**Summary Of The Review:**

I think the originality and evaluation of the paper need to be strengthened, including comprehensive comparison with state-of-the-art randomized algorithms as well as empirical evaluation on real-life data.

---

### Decision · Program_Chairs · 2022-01-20

**Decision:**

Reject

**Comment:**

The reviewers recommended rejection. There was no reply from the authors. The main weaknesses are:
- No experiment on real-life dataset (only simulated)
- Unsubstantiated claims about the literature
- No discussion on the time complexity
- Incremental contribution